# Distinctiveness Maximization in Datasets Assemblage

## Abstract

In this paper, given a user's query set and budget, we aim to use the limited budget to help users assemble a set of datasets that can enrich a base dataset by introducing the maximum number of distinct tuples (i.e., maximizing distinctiveness). We prove this problem to be NP-hard. A greedy algorithm using exact distinctiveness computation attains an approximation ratio of $(1 - e^{-1})/2$, but it lacks efficiency and scalability due to its frequent computation of the exact distinctiveness marginal gain of any candidate dataset for selection. This requires scanning through every tuple in candidate datasets and thus is unaffordable in practice. To overcome this limitation, we propose an efficient machine learning (ML)-based method for estimating the distinctiveness marginal gain of any candidate dataset. This effectively eliminates the need to test each tuple individually. Estimating the distinctiveness marginal gain of a dataset involves estimating the number of distinct tuples in the tuple sets returned by each query in a query set across multiple datasets. This can be viewed as the cardinality estimation for a query set on a set of datasets, and the proposed method is the first to tackle this cardinality estimation problem. This is a significant advancement over prior methods that were limited to single-query cardinality estimation on a single dataset and struggled with identifying overlaps among tuple sets returned by each query in a query set across multiple datasets. Extensive experiments using five real-world data pools demonstrate that our algorithm, which utilizes ML-based distinctiveness estimation, outperforms all relevant baselines in effectiveness, efficiency, and scalability. A case study on two downstream ML tasks also highlights its potential to find datasets with more useful tuples to enhance the performance of ML tasks.

## ACM Reference Format:

Anonymous Author(s). 2018. Distinctiveness Maximization in Datasets Assemblage. In *Proceedings of Make sure to enter the correct conference title from your rights confirmation emai (Conference acronym 'XX).* ACM, New York, NY, USA, 14 pages. https://doi.org/XXXXXXX.XXXXXXX

## 1 Introduction

Data is an essential resource for informed decision making [59]. This importance is reinforced by remarkable progress in machine learning (ML), which heavily relies on vast amounts of data to extract insights [50]. Hence, data preparation plays a pivotal role in transforming raw data into meaningful insights to support decision-making processes [70].

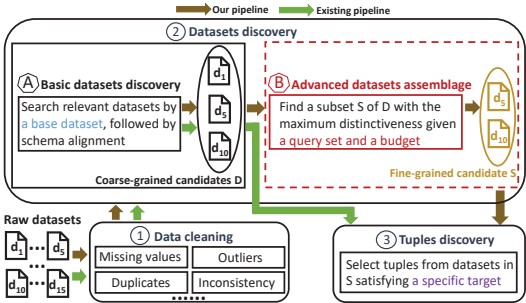

**Figure 1: Our data preparation pipeline with advanced datasets assemblage versus existing pipelines. The user input for each stage is shown by color (blue for basic datasets discovery, red for advanced datasets assemblage, and purple for tuples discovery).**

**A common pipeline of data preparation**. As shown in Fig. 1, it comprises three key stages: data cleaning, datasets discovery, and tuples discovery [50]. In Stage 1, cleaned datasets are obtained through data cleaning, involving tasks like missing value imputation [39] and duplicate removal [9]. In Stage 2, a subset of cleaned candidate datasets identified through datasets discovery is acquired by users to meet their information needs. At Stage 3, tuples are selected from the candidate datasets using tuples discovery to fulfill users' specific targets (e.g., enriching the training set of an existing ML model [7]). Despite extensive prior research efforts in datasets discovery (Stage 2) [2, 5, 6, 20, 22, 27, 44, 45, 58], an important research gap remains, limiting the practicality of existing solutions.

**The gap in the pipeline: datasets discovery**. Basic datasets discovery returns top-$k$ relevant datasets using keywords (e.g., AWS Marketplace [44]) or a base dataset (e.g., table union search [51]). Its search process generally evaluates each candidate dataset individually, focusing on the similarity or overlaps between individual datasets and given keywords or base datasets. This can bring substantial information redundancy when assembling these returned datasets [55]. Moreover, this approach *implicitly* assumes that users can afford all of the discovered datasets to later perform tuples discovery, which is often unrealistic. Users often have a limited budget and emphasize the return on investment [1].

**Our study: advanced datasets assemblage**. To bridge the above gap, we delve into advanced datasets assemblage. It builds upon the results of basic datasets discovery using a user's base dataset. It allows a user to specify her *fine-grained information needs* to assemble a useful dataset collection within a budget, aiming to evaluate the dataset collection as a whole to reduce information redundancy. Since SQL queries are widely used to pinpoint fine-grained and precise information within datasets [17, 38, 61], we employ a query set with SQL queries to allow a user to express her fine-grained information needs. Additionally, the schemas of candidate datasets obtained through basic datasets discovery can be *easily and effectively aligned* with the user's base dataset using the state-of-the-art schema alignment techniques, achieving high accuracy [53]. Therefore, the user can readily formulate her query set

**$d_5$**

| Type | City | Price |
|---|---|---|
| House | Melbourne | 520,000 |
| Apartment | Sydney | 735,000 |
| House | Sydney | 815,000 |
| House | Melbourne | 510,000 |

| Type | City | Price |
|---|---|---|
| Apartment | Sydney | 705,000 |
| House | Melbourne | 520,000 |
| House | Sydney | 970,000 |
| House | Melbourne | 750,000 |

**$d_{10}$**

**A query set Q**
$q_1$: SELECT * FROM d WHERE Type = 'House' and City = 'Melbourne'
$q_2$: SELECT * FROM d WHERE 500,000 < Price < 800,000 and City = 'Melbourne'

| House | Melbourne | 520,000 |
|---|---|---|
| House | Melbourne | 510,000 |

**$Q(d_5)$**

| House | Melbourne | 520,000 |
|---|---|---|
| House | Melbourne | 750,000 |

**$Q(d_{10})$**

| House | Melbourne | 520,000 |
|---|---|---|
| House | Melbourne | 510,000 |
| House | Melbourne | 750,000 |

**$Q(d_5) \cup Q(d_{10})$**

**Figure 2: An example for MCE (red for the overlapping tuple).**

using the schema information from her base dataset. We introduce the concept of *distinctiveness* to evaluate the usefulness of datasets discovered w.r.t. a user's fine-grained information needs. Let $Q(d)$ be the union of tuple sets returned by each query in a user's query set $Q$ on a dataset $d$ and $d_u$ be a user's base dataset. We define the distinctiveness for a dataset as *the size of $Q(d)$, including tuples in $Q(d_u)$*. Correspondingly, the distinctiveness for a set of datasets is defined as *the size of the union of $Q(d)$ over each dataset $d$ in a set of datasets, including tuples in $Q(d_u)$*. The usefulness of datasets is application-dependent and remains an open problem, but this distinctiveness definition serves as a starting point for assembling a useful dataset collection with distinct tuples as no one prefers to purchase an "assembled" dataset full of duplicate information. By maximizing distinctiveness, advanced datasets assemblage identifies datasets with minimal information redundancy within a budget, thereby locating useful datasets in a cost-effective manner.

To this end, we introduce the problem of *distinctiveness maximization in datasets assemblage*. Given a user's fine-grained information needs expressed as a query set, along with her base dataset and budget, and a set of candidate datasets discovered in basic datasets discovery, our goal is to select a subset of candidate datasets for user acquisition. This selection process maximizes the total distinctiveness for the subset within the user's budget. We exemplify this problem within the data preparation pipeline in Appendix A.

**The gap in a solution backbone**. We establish the NP-hardness of obtaining an exact solution for the distinctness maximization problem (See Appendix B), as well as showing a greedy algorithm using exact distinctiveness computation (Exact-Greedy for brevity) that can achieve an approximation ratio of $(1 − 1/e)/2$ in Appendix C.

However, the Exact-Greedy algorithm heavily relies on the frequent computation of the exact distinctiveness marginal gain of any dataset in candidate datasets during selection. The exact distinctiveness marginal gain of a dataset is computed as the difference between the distinctiveness for a set of datasets including the dataset and the distinctiveness for the dataset itself. Thus, the union $Q(d)$ of tuple sets returned by each query in a query set $Q$ over any dataset $d$ in candidate datasets must be obtained, which demands inspecting every tuple in the candidate datasets returned for the queries provided, significantly degrading both efficiency and scalability. To overcome this limitation, a natural choice is to effectively *approximate* the distinctiveness marginal gain rather than relying on the exact computation.

*From distinctiveness to cardinality estimation*. The key to estimating the distinctiveness marginal gain of a dataset is to estimate the distinctiveness for a set of datasets w.r.t. a query set $Q$. This involves estimating the size of the union of $Q(d)$ over each dataset $d$ in a set of datasets, and we refer to it as multi-dataset-query cardinality estim

ation (MCE). MCE can be viewed as a generalized version of the classical cardinality estimation problem [24, 26, 32, 42, 71, 76], which *estimates the cardinality of a single query for a single dataset* (i.e., the size of the tuple set returned by a query on a dataset). To distinguish between these two concepts, we refer to the latter as single-dataset-query cardinality estimation (SCE). As illustrated in Example 1.1 below, existing SCE solutions cannot be used to solve the MCE problem since they cannot capture both overlaps among the tuple sets returned by different queries on a dataset and overlaps among different datasets.

*Example 1.1 (Using the SCE solution for the MCE problem).* Consider the datasets $d_5$ and $d_{10}$, and a query set $Q = \{q_1, q_2\}$ shown in Fig. 2. Using an SCE solution, for $d_5$, the cardinality of $q_1$ is 2, and the cardinality of $q_2$ is 2. For $d_{10}$, the cardinality of $q_1$ is 2, and the cardinality of $q_2$ is 2. Since the SCE solution only reports the cardinality of each query for each dataset, the size of the union of $Q(d_5)$ and $Q(d_{10})$ is estimated to be 8, by aggregating the cardinality of each query on each dataset, which is much greater than the true result (three distinct tuples in the union of $Q(d_5)$ and $Q(d_{10})$).

**A novel solution**. To address this, we propose a novel ML-based method to estimate the distinctiveness for a set of datasets w.r.t. a query set. Specifically, we leverage a pre-trained model to transform the data summary of a dataset into embeddings that capture nuanced information from each dataset, queries, and their interrelationships to effectively *identify overlaps among the tuple sets returned by different queries on a dataset*. We use the embeddings created to estimate distinctiveness for the respective dataset. Furthermore, we incorporate a learning function to consolidate the data summaries from individual datasets to generate a data summary corresponding to a collection of the considered datasets while *identifying overlaps among datasets*. This allows us to estimate the distinctiveness of a set of datasets utilizing pertinent pre-trained models. Finally, we propose a new greedy algorithm that uses our ML-based distinctiveness estimation method (ML-Greedy for brevity) to address the distinctiveness maximization problem (§3).

**Evaluation**. We conduct extensive evaluations on five real-world data pools showcasing: 1) Our ML-based distinctiveness estimation method significantly outperforms the SOTA SCE solution on the MCE problem, with one-order-of-magnitude higher accuracy for estimating distinctiveness and is several times more efficient than the SOTA SCE solution. 2) Our ML-Greedy algorithm is competitive with Exact-Greedy and significantly outperforms other baselines for distinctiveness maximization. 3) Our ML-Greedy algorithm achieves impressive efficiency gains, with up to four orders of magnitude speedup over Exact-Greedy and three orders of magnitude improvement over the most efficient baseline method.

We conduct a case study on two downstream ML tasks, classification and regression, to assess the impact of our datasets assemblage methods, ML-Greedy and Exact-Greedy, on downstream task performance through the pipeline in Fig. 1. By comparing with the SOTA basic datasets discovery method, we validate our methods' potential in identifying datasets with more useful tuples(§4).

## 2 Problem Formulation

In this section, we outline our problem formulation and present the hardness analysis for our problem.

Given a user's base dataset $d_u$, a set $D = \{d_1, ..., d_{|D|}\}$ of candidate datasets is retrieved using basic datasets discovery. Each dataset $d \in D$ has a price $p(d) \in \mathbb{R}^+$, determined by a pricing function $p$. It is noteworthy that data pricing is an open problem and falls outside the scope of our study. We briefly discuss it in §5 (Related Work). We define the total price of $D$ as $p(D) = \sum_{d \in D} p(d)$. A user submits a query set with SQL queries $Q = \{q_1, ..., q_{|Q|}\}$ to specify fine-grained information needs. An SQL query can be expressed as SELECT * FROM $d$ WHERE ... AND $l_c \leq c \leq u_c$ AND ... where $d$ is a dataset and $c$ is a query column, handling both numerical and categorical data. For categorical data, $l_c$ equals $u_c$. The queries $q_1$ and $q_2$ in Fig. 2 are two examples.

A user interested in acquiring datasets may want to determine the number of unique tuples returned from a set $D$ of datasets based on a query set $Q$ and base dataset $d_u$. We refer to this concept as the *distinctiveness* of $D$ in relation to a user's information needs.

*Definition 2.1 (Distinctiveness).* Suppose $q(d)$ is the tuple set returned by applying query $q$ to dataset $d$, and $Q(d)$ is the union of the tuple sets returned by each $q \in Q$ on $d$, referred to as $Q(d) = \cup_{q \in Q} q(d)$. The *distinctiveness* $\mathscr{D}(S, d_u, Q)$ of a set of datasets $S$ is the size of the union of $Q(d)$ over all $d \in S \cup d_u$. That is, $\mathscr{D}(S, d_u, Q) = \left| \bigcup_{d \in S \cup d_u} Q(d) \right|$.

As mentioned in §1, the distinctiveness estimation problem is analogous to the MCE problem, as formalized below.

*Definition 2.2 (Multi-dataset-query cardinality estimation (MCE)).* Given a set $D$ of datasets and a query set $Q$, MCE estimates the cardinality of $Q$ for $D$, which is the size of the union of $Q(d)$ over each dataset $d \in D$, i.e., $\left| \bigcup_{d \in D} Q(d) \right|$

In the rest of this paper, we use the concepts of distinctiveness estimation and MCE interchangeably. Notably, when $Q$ has only one query ($|Q| = 1$) and $D$ contains only one dataset ($|D| = 1$), the distinctiveness estimation problem is equivalent to the well-known SCE problem [42, 71, 76].

*Definition 2.3 (Distinctiveness Maximization (DM)).* Given a budget $B$, a query set $Q$, a base dataset $d_u$, a set $D$ of candidate datasets, and a pricing function $p$, the DM problem returns a subset $S^* \subseteq D$ that yields the maximum distinctiveness within the budget $B$. Formally, we have

$$S^* = \operatorname{argmax}_{S \subseteq D} \mathscr{D}(S, d_u, Q) \text{ s.t. } \sum_{d \in S} p(d) \leq B. \tag{1}$$

We prove the NP-hardness of the DM problem using a reduction from the maximum coverage (MC) problem [48] in Appendix B.

## 3 Greedy Algorithm using ML-based Distinctiveness Estimation

Given the NP-hardness of the DM problem, obtaining an optimal solution is computationally intractable, even for moderately sized datasets. To this end, a straightforward choice is to apply a greedy algorithm to find an approximate solution (Alg. 1 shows a skeleton of the greedy algorithm). We prove that a greedy algorithm using exact distinctiveness computation (Exact-Greedy for brevity) achieves an approximation ratio of $(1 - e^{-1})/2$. We present more details of Exact-Greedy in Appendix C.

However, Exact-Greedy is still computationally expensive due to *the exact computation of the distinctiveness marginal gain* (i.e.,

---

**Algorithm 1** The Greedy Skeleton

**Input:** a set $D$ of datasets, a base dataset $d_u$, a query set $Q$, a budget $B$;
**Output:** a subset $S \subseteq D$ of datasets with distinctiveness;
1: $S \leftarrow \emptyset$, $\mathscr{D}(S, d_u, Q) \leftarrow 0$
2: **while** $D \neq \emptyset$ **do**
3:     $d^* \leftarrow \arg\max_{d \in D} \mathscr{D}(S \cup d, d_u, Q) - \mathscr{D}(S, d_u, Q)$;
4:     **if** $p(S) + p(d^*) \leq B$ **then** $S \leftarrow S \cup d^*$, update $\mathscr{D}(S, d_u, Q)$;
5:     $D \leftarrow D \setminus d^*$;
6: $d_t \leftarrow \arg\max_{d \in D \wedge p(d) \leq B} \mathscr{D}(\{d_t\}, d_u, Q)$;
7: **if** $\mathscr{D}(S, d_u, Q) < \mathscr{D}(\{d_t\}, d_u, Q)$ **return** $\{d_t\}$ and $\mathscr{D}(\{d_t\}, d_u, Q)$;
8: **return** $S$ and $\mathscr{D}(S, d_u, Q)$;

---

$\mathscr{D}(S \cup d, d_u, Q) - \mathscr{D}(S, d_u, Q))$. This is not practical since a data preparation system in production must deliver results to users quickly in order to avoid user abandonment. Therefore, a natural question arises: Is it possible to efficiently estimate the distinctiveness and maintain high effectiveness? As aforementioned in §1, distinctiveness estimation is analogous to the MCE problem, with the goal of estimating the cardinality of a query set on a set of datasets. It cannot be addressed by a solution of the SCE problem [42]. Thus, we devise a new ML-based method for solving the MCE problem.

**Overview for our ML-based distinctiveness estimation method.** As shown in Fig. 3, our method consists of five components.

In an offline process, we use Component 1 to generate a data summary for each dataset under consideration. In an online process, we use the data summary from each dataset and a query set as the input. Using Components 2-4, we estimate the distinctiveness for a dataset w.r.t. a query set while identifying overlaps among the tuple sets returned by each query in a query set on a dataset. Specifically, we first utilize Component 2 to transform the data summary into a query-aware dataset embedding, which captures the connections between the query set and the data summary. Then, we use Component 3 to generate a query-set embedding by merging the information from the queries in the query set. Finally, Component 4 estimates the distinctiveness for a dataset w.r.t. a query set using the embeddings generated in the previous components. To estimate the distinctiveness for a set of datasets w.r.t. a query set using the above components, we first compute a data summary for a set of datasets by identifying overlaps among them. The data summary for a set of datasets is computed by iteratively merging the current data summary with the data summary of a new dataset, starting with the first dataset in the set. To this end, our ML-based method uses Component 5, which includes a learning function, to efficiently generate a data summary for a set of datasets by merging the data summaries for individual datasets.

### 3.1 Component 1: Data Summary Generation

We adopt the pre-training summarization approach IRIS [42] to generate data summaries (see the left side of Fig. 3). When compared to other possible approaches such as histograms [29] or sketches [13], IRIS does not require per-dataset training. Hence, it reduces the total cost required to generate the summaries while producing high-quality results for the SCE problem.

Specifically, IRIS has a two-step process. The first step is *identifying column sets* where a set $C_d$ of columns, which are highly correlated with dataset $d$, is located. The second step, *generating column set embeddings*, uses pre-trained models to create an embedding $e_C$ for each column set $C \in C_d$. These column set embeddings collectively form a data summary $E_C^d = \{..., e_{C_i}, ...\}$ where $C_i \in C_d$. Further details on these two steps are provided below.

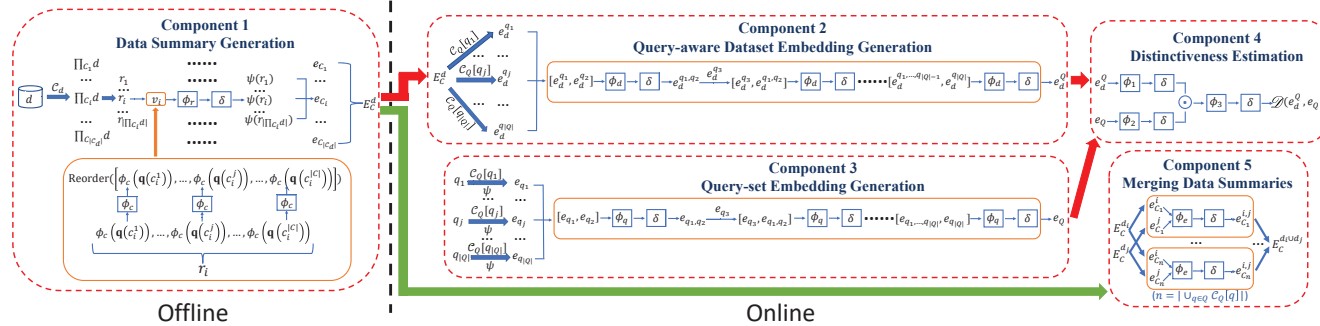

**Figure 3: The ML-based distinctiveness estimation method, where red arrows represent the process of estimating the distinctiveness of a dataset and green arrows represent the process of merging data summaries.**

*3.1.1 Identifying column sets.* For each dataset $d \in D$, a small number of tuples is randomly selected from $d$ and the correlation scores for each column pair are computed using CORDS [28]. A graph is then constructed with columns as nodes and edges weighted using their correlation scores. Using this graph, a set of nodes (columns) is selected iteratively to form a column set. During each iteration, a clique, containing no more than $\kappa$ (2 in our experiments) nodes, with the highest total edge weight is selected as a column set. Then, all edges from the chosen clique are removed from the graph.

Let $\Pi_C d$ denote a projection of dataset $d$ on columns in $C$. The process of selecting column sets continues until the combined storage size of $\Pi_C d$ of all of the column sets reaches a pre-defined limit (4 kb in our experiments), resulting in a set of column sets $C_d$, e.g., in Fig. 2, $C_{d_1}$ of $d_5$ is $\{C_1 = \{\text{Type}, \text{City}\}, C_2 = \{\text{City}, \text{Price}\}\}$.

*3.1.2 Generating column set embeddings.* For each column set $C \in C_d$ and corresponding rows $\Pi_C d$ from dataset $d \in D$, a row embedding is learned for each row $r \in \Pi_C d$. Then, the average for all of the row embeddings for $C$ is computed, which is the column set embedding $e_C$ of $C$, i.e,

$$e_C \triangleq \frac{1}{|\Pi_C d|} \sum_{i=1}^{|\Pi_C d|} \psi(r_i). \tag{2}$$

Here, $\psi(\cdot)$ is a learned function (model) for a row embedding, computed using a quantization function $\mathbf{q}(\cdot)$ [42], a column embedding function $\phi_c(\cdot)$, a row embedding function $\phi_r(\cdot)$, and a ReLU activation function $\delta(\cdot)$ as

$$\psi(r_i) \triangleq \delta(\phi_r(\text{Reorder}([\cdots, \phi_c(\mathbf{q}(c_i^j)), \cdots]))), \tag{3}$$

where $\phi_c : \mathbb{R}^\xi \to \mathbb{R}^{64}$, $\phi_r : \mathbb{R}^\ell \to \mathbb{R}^\eta$, and $c_i^j$ are values of the $j$-th column in $C$ for row $r_i$. The Reorder function reorders the values based on the total number of bits assigned, such that the most distinct columns are aligned with the same input position for $\phi_r$. A quantization function $\mathbf{q}(\cdot)$ replaces the value for each column, with an identifier containing at most $\xi$ bits, while $\phi_c(\mathbf{q}(c_i^j))$ encodes an identifier $\mathbf{q}(c_i^j)$ in an embedding layer. Let $v_i$ be a vector concatenating the reordered outputs for $\phi_c$. The function $\phi_r(v_i)$ creates a row embedding of $r_i \in \Pi_C d$ using a fully-connected neural-network (NN) layer. The function $\phi_r(v_i)$ receives $\ell$ bits as input and outputs $\eta$ bits (the size of the row embedding). Finally, $\delta(\phi_r(v_i))$ is the ReLU activation function is applied to $\phi_r(v_i)$. Using Eq. 2, a column set embedding $e_C$ of $C$ is computed with size $\eta$.

*Example 3.1.* Consider the column set $C = \{\text{City}, \text{Price}\}$ from dataset $d_5$ in Fig. 2. We allocate $\ell = 3$ bits for the two columns, containing 2 and 4 distinct values, respectively. Using a quantization

approach [42], we assign $[1, 2]$ bits to the columns. Using the embedding method, we produce $\phi_c(\mathbf{q}(c_1^1)) = [0]$, $\phi_c(\mathbf{q}(c_1^2)) = [0, 1]$, and $\text{Reorder}[\phi_c(\mathbf{q}(c_1^1)), \phi_c(\mathbf{q}(c_1^2))] = [\phi_c(\mathbf{q}(c_1^2)), \phi_c(\mathbf{q}(c_1^1))] = [0, 1, 0]$ for $r_1 \in \Pi_C d$. Thus, we have $\psi(r_1) = \delta(\phi_r([0, 1, 0]))$.

## 3.2 Component 2: Query-aware Dataset Embedding Generation

To effectively represent any relationships between the queries in $Q$ and a data summary $E_C^d$ of $d$, we create a query-aware dataset embedding $e_d^Q$ using a learned model $\phi_d(\cdot)$. This process is shown in the "query-aware dataset embedding generation" portion of Fig. 3.

To improve overall efficiency, we construct a lookup table $C_Q$ to store the column sets linked to each query $q \in Q$, i.e., $C_Q[q] = \{C | C \in \cap_{d \in D} C_d \ \& \ C \cap \text{ColsOf}(q)\}$. Here, $\text{ColsOf}(q)$ refers to the query columns. Each column set in $C_q$ is also a column set for a dataset in $D$. For a given data summary $E_C^d$ for dataset $d$ and lookup table $C_Q[q]$, we can construct the dataset embedding $e_d^q$ for each query $q \in Q$. To this end, we concatenate all column set embeddings from $E_C^d$ which are associated with $C_Q[q]$. This produces $e_d^q = [..., e_{C_i}, ...]$, where $e_{C_i} \in E_C^d$ and $C_i \in C_Q[q]$.

Next, we construct $\phi_d : \mathbb{R}^{2\eta x} \to \mathbb{R}^{\eta x}$, which is a fully-connected layer with ReLU activation function $\delta(\cdot)$. It is designed to iteratively learn a query-aware dataset embedding for a dataset $d$, i.e.,

$$e_d^{q_1,q_2} = \delta(\phi_d([e_d^{q_1}, e_d^{q_2}])) \dots e_d^{q_1,\dots,q_j} = \delta(\phi_d([e_d^{q_j}, e_d^{q_1,\dots,q_{j-1}}]))$$
$$\dots e_d^Q = \delta(\phi_d([e_d^{q_{|Q|}}, e_d^{q_1,\dots,q_{|Q|-1}}])). \tag{4}$$

For the first two queries, $q_1, q_2 \in Q$, we concatenate the respective dataset embeddings, $e_d^{q_1}$ and $e_d^{q_2}$, as the input for $\phi_d$. The resulting embedding, $e_d^{q_1,q_2}$, combines the information of both $e_d^{q_1}$ and $e_d^{q_2}$. Similarly, we can generate the embedding $e_d^{q_1,\dots,q_j}$ by combining $e_d^{q_j}$ with $e_d^{q_1,\dots,q_{j-1}}$. Following this process, a final query-aware dataset embedding $e_d^Q = e_d^{q_1,\dots,q_{|Q|}}$ is created. Since the size of each column set embedding is $\eta$, the input size for $\phi_d(\cdot)$ is a constant, $2\eta x$, and the output size is $\eta x$. In cases where the input size is less than $2\eta x$, zero-padding can be applied to achieve the required size.

## 3.3 Component 3: Query-set Embedding Generation

In this section, we describe how to generate a query-set embedding $e_Q$ which merges information for all queries in a query set $Q$, as shown in the "query-set embedding generation" portion of Fig. 3.

For each query $q \in Q$, $C \in C_Q[q]$ represents the column set included in the WHERE clause from $q$. As discussed in §2, each column $c \in C$ is associated with a query range defined by the minimum value $l_c$ and maximum value $u_c$. We can represent a query as $q = [q_l^C, q_h^C]$, where $q_l^C = [l_{c_1}, ..., l_{c_{|C|}}]$ and $q_h^C = [u_{c_1}, ..., u_{c_{|C|}}]$. To generate the embeddings for a query $q$ relative to the set $C$, we use the same row embedding function $\psi(\cdot)$ (see Eq. 3) for both the minimal and maximal values of each column, namely $q_l^C$ and $q_h^C$. This produces two distinct embeddings for $q$ in $C$, $\psi(q_l^C)$ and $\psi(q_h^C)$. Next, a complete query embedding $e_q$ is created for $q$ by concatenating all of the embeddings for $q$, for each column set $C \in C_Q[q]$, i.e., $e_q = [..., \psi(q_l^{C_i}), \psi(q_h^{C_i}), ...]$ where $C_i \in C_Q[q]$.

*Example 3.2.* Consider the column set $C$ = {City, Price} in Example 3.1 and the query $q_2$ = SELECT Count(*) FROM d WHERE 500,000 ≤ Price ≤ 800,000 AND City = 'Melbourne'. We have $\phi_c(\mathbf{q}(\text{Melbourne})) = [0]$ and $\phi_c(\mathbf{q}(\text{Sydney})) = [1]$, which is the range of all distinct values in City. We also have $\phi_c(\mathbf{q}(500,000)) = [0,0]$ and $\phi_c(\mathbf{q}(800,000)) = [1,0]$. Therefore, Reorder$[\phi_c(\mathbf{q}(\text{Melbourne})), \phi_c(\mathbf{q}(500,000))] = [0,0,0]$ and Reorder$[\phi_c(\mathbf{q}(\text{Sydney})), \phi_c(\mathbf{q}(800,000))] = [1,0,1]$. This results in the query embedding $[\psi(q_l^C), \psi(q_h^C)] = [\delta(\phi_r([0,0,0])), \delta(\phi_r(1,0,1))]$ representing $q_2$.

After generating all of the individual query embeddings, we apply an iterative procedure as described in §3.2 to produce the final query-set embedding as follows:

$$e_{q_1,q_2} = \delta(\phi_p([e_{q_1}, e_{q_2}])) \dots e_{q_1,...,q_j} = \delta(\phi_p([e_{q_j}, e_{q_1,...,q_{j-1}}]))$$
$$\dots e_Q = \delta(\phi_p([e_{q_{|Q|}}, e_{q_1,...,q_{|Q|-1}}])). \tag{5}$$

Here, $\phi_q : \mathbb{R}^{4\eta x} \to \mathbb{R}^{2\eta x}$ is a fully-connected layer and $\delta(\cdot)$ is the ReLU activation function. Since the size of the column set embedding for a query is $2\eta$, we fix the input size of $\phi_q(\cdot)$ to $4\eta x$ and the output size of $\phi_q(\cdot)$ is $2\eta x$. In cases where the input size is less than $4\eta x$, zero-padding is applied to the embedding to ensure that the input size is equal to $4\eta x$.

## 3.4 Component 4: Distinctiveness Estimation

Using our query-aware dataset embedding $e_d^Q$ and query-set embedding $e_Q$ as inputs, we develop three learned models, namely, $\phi_1(\cdot)$, $\phi_2(\cdot)$, and $\phi_3(\cdot)$, which allow us to estimate the distinctiveness of each dataset $d$ as

$$\mathscr{D}(e_d^Q, e_Q) = \delta(\phi_3(\delta(\phi_1(e_d^Q)) \odot \delta(\phi_2(e_Q)))). \tag{6}$$

Here, $\phi_1 : \mathbb{R}^{\eta x} \to \mathbb{R}^{\eta}$ and $\phi_2 : \mathbb{R}^{2\eta x} \to \mathbb{R}^{\eta}$ are two fully-connected layers and $\phi_3 : \mathbb{R}^{\eta} \to \mathbb{R}^1$ is a multilayer perceptron with one hidden layer. $\delta(\cdot)$ is the ReLU activation function.

**The distinctiveness estimation process**. As shown in Fig. 3, our distinctiveness estimation process relies on Components 1 - 4, each of which includes one or more models. Specifically, $\phi_c$ and $\phi_r$ are used to generate the data summary, $\phi_d$ is used to generate the query-aware dataset embedding, $\phi_q$ is used to generate the query-set embedding, and $\phi_1$, $\phi_2$, and $\phi_3$ are used to estimate distinctiveness. These models are end-to-end pre-trained [23]. The pseudocode for distinctiveness estimation is presented in Appendix D. First, we compute a dataset embedding $e_d^q$ for each query $q \in Q$ together with a query embedding $e_q$ for each query $q \in Q$ through $C_Q[q]$ and $E_C^d$. Next, we generate the query-aware dataset

embedding $e_d^Q$ followed by the query-set embedding $e_Q$. Finally, we estimate the distinctiveness of dataset $d$. Note that while we adopt an iterative approach to generate both query-aware dataset and query-set embeddings, the order in which the queries are processed minimally affects the final results.

**The loss function**. Consider a set $\mathcal{D}$ of training instances where each instance includes an estimate for distinctiveness $\widetilde{\mathscr{D}}(d, Q)$ for $d$ given $Q$ as well as the exact value of distinctiveness $\mathscr{D}(d, Q)$. The mean squared error (MSE) loss is defined as $\frac{1}{|\mathcal{D}|} \sum_{(\widetilde{\mathscr{D}}(d,Q), \mathscr{D}(d,Q)) \in \mathcal{D}} (\widetilde{\mathscr{D}}(d,Q) - \mathscr{D}(d,Q))^2$.

## 3.5 Component 5: Merging Data Summaries

This component is designed to compute a data summary $E_C^{S \cup d}$ for $S \cup d$ by merging the data summary $E_C^S$ for the current set of datasets, $S$, and the data summary $E_C^d$ for a dataset $d$. The objective is to estimate the distinctiveness of $S \cup d$ w.r.t $Q$ using Alg. 3.

IRIS [42] incrementally updates the data summary $E_C^d$ for the dataset $d$ when new rows are added to $d$. Specifically, for each column set $C$ of $d$, with new rows in $C$ denoted as $R_{\text{new}}$, a new column set embedding is computed as $e_C' = (ne_C + \sum_{r \in R_{new}} \psi(r))/(n + |R_{new}|)$. Here, $e_C$ is the column set embedding for $C$ in $E_C^d$ and $\psi(\cdot)$ is the row embedding function from Eq. 3. The column set embedding $e_C$ in $E_C^d$ is then updated to $e_C'$.

However, this method is computationally expensive when updating the data summary $E_C^S$ for the current set $S$ of datasets with new rows from a dataset $d$. So to better facilitate the update process, it is necessary to check if any existing row in $d$ is already included in $S$. In such a scenario, a new model must be trained to learn the column set embeddings for the data summary. This process is summarized in the "merging data summaries" portion of Fig. 3. Given two datasets $d_i$ and $d_j$, $d_{i,j}$ is the dataset resulting from the merge process. Let $e_C^i$ be a column set embedding for $d_i$, $e_C^j$ for $d_j$, and $e_C^{i,j}$ for $d_{i,j}$, where $C \in \cup_{q \in Q} C_Q[q]$. The embedding is learned using $e_C^i$ and $e_C^j$, which approximates $e_C^{i,j}$ using:

$$\delta(\phi_e([e_C^i, e_C^j])) \approx e_C^{i,j} \tag{7}$$

where $\phi_e : \mathbb{R}^{2\eta} \to \mathbb{R}^{\eta}$ is a fully-connected layer.

**Merging data summaries**. The procedure for merging data summaries is shown in Appendix D. Given a data summary $E_C^d$ for the dataset $d$, a data summary $E_C^S$ for the set of datasets $S$, and a lookup table $C_Q$, which maps column sets $C_q$ associated with each query $q \in Q$, using Eq. 7 to update column set embeddings in $E_C^S$ whose column set is in $\cup_{q \in Q} C_q$.

**The loss function**. Consider a set $\mathcal{S}$ of training instances where each instance is the pair of column embedding $\widetilde{e}_C^{i,j}$ for the column set $C$ on the dataset created by merging $d_i$ and $d_j$ and the corresponding ground-truth column pair embedding $e_C^{i,j}$. The MSE loss is $\frac{1}{|\mathcal{S}|} \sum_{(\widetilde{e}_C^{i,j}, e_C^{i,j}) \in \mathcal{S}} (\widetilde{e}_C^{i,j} - e_C^{i,j})^2$.

## 3.6 The Complete Algorithm

Combining all components in §3.1-3.5, we present ML-Greedy, a greedy algorithm using ML-based distinctiveness estimation. The process begins offline, generating the set of column sets $C_d$ and

**Table 1: q-error of DE and IRIS for varying percentiles.**

| Dataset | Method | q-error | | | | |
|---------|--------|------|------|------|------|-------|
| | | 0th | 25th | 50th | 75th | 100th |
| TPCH | DE | 1.602 | 1.661 | 1.669 | 1.682 | 1.801 |
| | IRIS | 11.629 | 11.713 | 11.938 | 11.981 | 12.423 |
| DMV | DE | 1.139 | 1.171 | 1.189 | 1.212 | 1.237 |
| | IRIS | 12.105 | 12.145 | 12.173 | 12.456 | 12.658 |
| IMDB | DE | 1.69 | 1.694 | 1.701 | 1.705 | 1.762 |
| | IRIS | 12.17 | 12.229 | 12.245 | 12.276 | 12.322 |
| Airline | DE | 1.725 | 1.840 | 1.85 | 1.858 | 1.876 |
| | IRIS | 12.07 | 12.087 | 12.103 | 12.134 | 12.18 |
| RealEstate | DE | 1.579 | 1.643 | 1.673 | 1.706 | 1.756 |
| | IRIS | 12.186 | 12.22 | 12.251 | 12.313 | 12.445 |

the data summary $E_C^d$ for each dataset $d \in D$ and the base dataset $d_u$. Next, an online process is executed. Initially, a lookup table $C_Q$ is created to store column sets associated with each query $q \in Q$. From the outset, the data summary $E_C^S$ of the current set of datasets $S$ is set to the data summary $E_C^{d_u}$ of $d_u$ and has the distinctiveness of $d_u$. In the first iteration, the distinctiveness $\mathscr{D}(\{d\}, d_u, Q)$ for each individual dataset $d$ is computed using our distinctiveness estimation process in §3.4 and recorded in $\mathbb{D}$. In subsequent iterations, the marginal gain $g = \mathscr{D}(S \cup d, d_u, Q) - \mathscr{D}(S, d_u, Q)$ when adding $d$ to the set $S$ is computed after the data summaries for $S$ and $d$ are merged using our method for merging data summaries in §3.5. The set $S$ is then iteratively updated by adding the dataset $d^*$ that yields the highest marginal gain $g^*$ until the budget is exceeded. Finally, we locate the single dataset $d_t$ with the maximum distinctiveness. It selects a better solution from the datasets in $S$ and a single dataset $d_t$. The pseudo-code of ML-Greedy and its time complexity analysis are shown in Appendix D.

## 4 Experiments

We conduct extensive experiments to verify the effectiveness, efficiency, and scalability of our greedy algorithm using ML-based distinctiveness estimation.

### 4.1 Experimental Setup

**Preparing datasets and queries**. Since real-world user queries and datasets are typically only available to commercial data marketplaces, we propose several strategies to carefully prepare candidate datasets and query sets using five real-world data pools of varying sizes and column types. The key behind these strategies is to control *overlapping tuples in the candidate datasets* and *overlapping tuples in the tuple sets returned by different queries*. To achieve this, we introduce the following parameters for preparing datasets and queries: (1) $s_{min}$ and $s_{max}$ set the sampling rate lower and upper bounds for each candidate dataset (For simplification, we fix $s_{min}$ in our experiments); (2) $ol$ controls the minimum overlap ratio in the tuples returned by any pair of queries for a dataset. We believe this helps reflect various (extreme) scenarios in real world. Please refer to Appendix E for more details.

**Methods for comparison**. Our ML-based distinctiveness estimation method includes two important components (1) distinctiveness estimation and (2) updating data summaries. To demonstrate the effectiveness of our method, we explore several alternatives for both components. For component (1), we use two methods: DE – our distinctiveness estimation method as described in §3.4; IRIS – A state-of-the-art solution for the SCE problem [42], where the

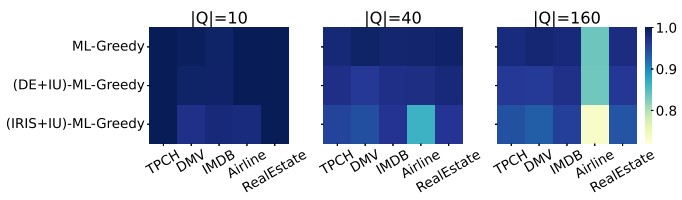

**Figure 4: The impact of the number of queries $|Q|$ on the distinctiveness ratio of each algorithm.**

results can be aggregated for all queries to approximate the distinctiveness of a query set. For component (2), we also consider two methods: MS – our approach to merge data summaries, as described in §3.5; IU – A method that identifies new tuples from a new dataset not present in the current set of datasets, then update column set embeddings using the new tuples, as outlined in §3.5. This results in four different approaches being compared:
- ML-Greedy – Our greedy algorithm using ML-based distinctiveness estimation, as described in §3.6.
- (DE+IU)-ML-Greedy – A greedy algorithm that uses DE for distinctiveness estimation and IU for updating data summaries.
- (IRIS+IU)-ML-Greedy – A greedy algorithm that uses IRIS for distinctiveness estimation and IU for updating data summaries.
- Exact-Greedy – The greedy algorithm using exact distinctiveness computation as described in Appendix C.

**Evaluation Metrics**. We use the following metrics:
- *Distinctiveness ratio* ($\mathscr{D}$-ratio) is the ratio of the estimated distinctiveness $\mathscr{D}$ to the exact distinctiveness $\mathscr{D}_{gt}$ calculated by Exact-Greedy, i.e, $\mathscr{D}$-ratio = $\mathscr{D}/\mathscr{D}_{gt}$.
- *Runtime*. The average time over five independent runs.

**Implementation**. Please see Appendix F.1 for specific parameter settings, training set generation, and experimental environment. All source code is publicly available [12].

### 4.2 Accuracy of Distinctiveness Estimation

We first verify the effectiveness of our distinctiveness estimation method DE. As discussed in §4.1, we have considered two different implementations – our distinctiveness estimation method DE and a state-of-the-art SCE method IRIS. We compare the effectiveness using q-error, which is widely used in prior works on cardinality estimation [32, 42]. Using the default settings, we estimate the distinctiveness for each dataset in $D$, and then compute the q-error for all of the datasets. Table 1 compares the performance when varying the percentiles of test cases. Observe that the q-error for DE is more than an order of magnitude less than IRIS. This implies that our distinctiveness estimation method is considerably more effective than the adapted SOTA SCE approach.

### 4.3 Effectiveness Study

ML-Greedy is compared to its variants and Exact-Greedy in Figs. 4-8. We observed: (1) The distinctiveness ratios of ML-Greedy and (DE+IU)-ML-Greedy can be up to 13% larger than (IRIS+IU)-ML-Greedy. Since $|d|$ is typically large in practice, such a difference – 13% – in distinctiveness ratio is equivalent to a difference of tens of thousands of tuples. This shows that our distinctiveness estimation method DE is effective and reliable. (2) The distinctiveness

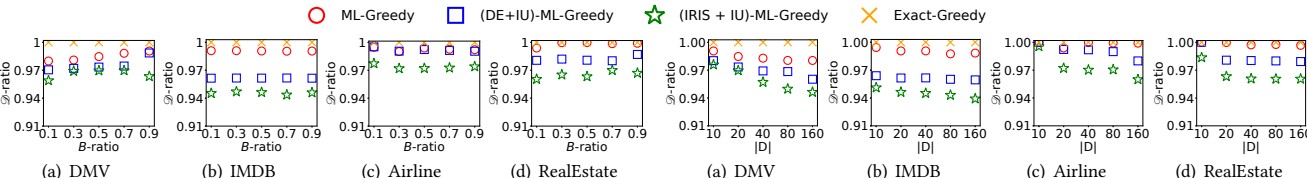

Figure 5: The impact of budget $B$ on the distinctiveness ratio of each algorithm. (no line chart since cases are independent)

Figure 6: The impact of the number of datasets $|D|$ on the distinctiveness ratio of each algorithm.

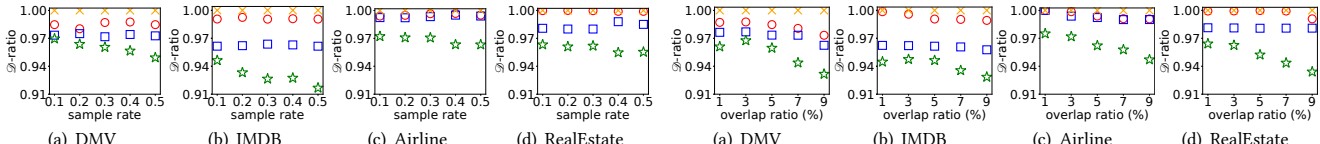

Figure 7: The impact of the sampling rate upper bound $s_{\max}$ on the distinctiveness ratio of each algorithm.

Figure 8: The impact of the minimum overlap ratio $ol$ between query pairs on the distinctiveness ratio of each algorithm.

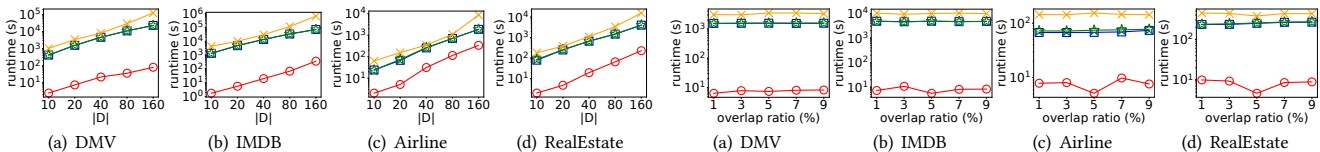

Figure 9: The impact of the number of datasets $|D|$ on the runtime of each algorithm.

Figure 10: The impact of the minimum overlap ratio $ol$ between query pairs on the runtime of each algorithm.

estimated by both `ML-Greedy` and `(DE+IU)-ML-Greedy` are competitive with those of `Exact-Greedy`, especially on smaller datasets such as Airline. (3) The distinctiveness ratio of `ML-Greedy` is consistently higher than `(DE+IU)-ML-Greedy`. This implies that our approach for updating data summaries (MS) is highly effective, regardless of the number of datasets (controlled by $|D|$) or the level of similarity between dataset distributions (controlled by $s_{max}$). (4) As $s_{max}$ grows, the distinctiveness ratio of `(IRIS+IU)-ML-Greedy` generally decreases. Since each dataset adds additional tuples as $s_{max}$ increases, the ability of IRIS to predict the number of tuples is worse than DE. Recall that distinctiveness is the product of Eq. 6 and the number of tuples in a dataset, so the error increases as the number of tuples gets larger. (5) The distinctiveness ratio of `(IRIS+IU)-ML-Greedy` decreases as $|Q|$ or $|D|$ increases as it ignores overlaps in query set results among datasets while a larger $|Q|$ or $|D|$ typically leads to more overlapping tuples. (6) In some cases, the distinctiveness ratios of `ML-Greedy` and `(DE+IU)-ML-Greedy` may also decrease with increasing $|Q|$, but both still perform well (> 83%), which is a testament to the merits of the newly proposed distinctiveness estimation method DE, in accounting for overlaps in the query set results.

## 4.4 Efficiency and Scalability Study

We present the runtime for `ML-Greedy`, along with its variants and `Exact-Greedy`. The impact of varying the number of datasets $|D|$ and the minimum overlap ratio $ol$ between query pairs is shown in Fig. 9 and Fig. 10, respectively. Please refer to Appendix F.3 for the impact of varying other parameters. We observe: (1) `ML-Greedy` is up to four orders of magnitude faster than all other algorithms. This translates to better scalability, especially when $|D|$ is large. (2) Our distinctiveness estimation method DE is up to three orders of magnitude faster than the exact distinctiveness computation, and is up to four times faster than IRIS. (3) Although `(DE+IU)-ML-Greedy`

Table 2: Comparison against basic datasets discovery where $S$ represents discovered datasets, the best model performance is in bold (RMSE for House, AUC for HR).

| Tasks | Methods | $\sum_{d \in S} |d|$ | $|\bigcup_{d \in S} d|$ | Model performance |
|---|---|---|---|---|
| | D3L | 56,750 | 52,472 | 0.601 |
| House | Exact-Greedy | 133,391 | 110,404 | **0.569** |
| | ML-Greedy | 133,391 | 110,404 | **0.569** |
| | D3L | 6711 | 3196 | 60.3% |
| HR | Exact-Greedy | 6270 | 5325 | 63.5% |
| | ML-Greedy | 6270 | 5325 | **63.5%** |

uses the same distinctiveness estimation method as `ML-Greedy`, it is up to two orders of magnitude slower than `ML-Greedy`. This demonstrates the high efficiency of our approach, MS, to update data summaries. (4) When $ol$ increases, the runtime for all of the algorithms has little dependence on $ol$. This indicates that the number of overlapping tuples returned by queries has minimal impact on runtime, even though our distinctiveness estimation method considers the overlaps.

## 4.5 Case Study

Now, we conduct a case study to test the hypothesis that our datasets assemblage can help find datasets with more useful tuples than basic datasets discovery. The point is that, we compare our datasets assemblage methods, `ML-Greedy` and `Exact-Greedy`, with a state-of-the-art basic datasets discovery method D3L [5], to show how `ML-Greedy`, `Exact-Greedy` and D3L impact downstream task performance through the pipeline illustrated in Fig. 1.

To this end, we consider two ML downstream tasks in [66], i.e., a classification task "predicting whether an employee will change their job" using a multilayer perception (MLP) model (House for brevity), and a regression task "predicting the price of a house" using a support vector regression (SVR) model (HR for brevity). For each ML task, the authors of [66] provide an initial training set $d_{train}$, a test set $d_{test}$, a validation set $d_{val}$, and a data pool

$d_{pool}$. Using $d_{pool}$, we create a set $D$ with 20 candidate datasets and a user's query set $Q$ with 10 queries following the procedure of Appendix E. We use $d_{train}$ as a user's base dataset $d_u$. Then, we address the following problems:

- *Basic datasets discovery*: given a user's base dataset $d_u$ and a set $D$ of candidate datasets, it aims to find top-$k$ datasets relevant to $d_u$.
- *Advanced datasets assemblage*: given a user's base dataset $d_u$, along with her query set $Q$ and her budget $B$, and a set $D$ of candidate datasets, it aims to find a subset of $D$ such that the subset has the maximum distinctiveness within the budget.
- *Tuples discovery*: The user's target, as in [66], is to enrich her base dataset for model training. Thus, given a user's base dataset $d_u$, along with her test set $d_{test}$, her validation set $d_{val}$ and her ML model $M$, and a data pool $P$ after assembling datasets discovered by basic datasets discovery or advanced datasets assemblage, it aims to select a set of tuples from $P$ to enrich $d_u$, such that the performance of $M$ has the maximum improvements.

Since D3L does not account for dataset prices, for a fair comparison, we set the price of each candidate dataset to 1 and $k = B = 5$. For tuples discovery, we apply the IAS algorithm in [66], stopping when the number of selected tuples is equal to the number of tuples in $d_u$. As shown in Table 2, `ML-Greedy` can find the same datasets as `Exact-Greedy` for both ML tasks, showing that our ML-based distinctiveness estimation is very close to exact distinctiveness computation. Our datasets assemblage methods, `ML-Greedy` and `Exact-Greedy`, can identify more distinct tuples than D3L, even when the total number of tuples in the datasets they discover is smaller than in those discovered by D3L. For both ML tasks, models trained on datasets enriched by our datasets assemblage methods perform better. This suggests that our datasets assemblage methods' potential in finding those datasets with more useful tuples, as a higher number of distinct tuples leads to more diverse data distributions in the training set, resulting in better model training [37].

## 5 Related Work

**Datasets discovery** can be broadly divided into basic datasets discovery and advanced datasets assemblage.

*Basic datasets discovery* is normally formulated as a search problem [8]. It relies on keywords [6, 44] or a base dataset [5, 20, 22, 27, 58] to retrieve relevant datasets. For example, table union search [20, 22, 27, 58] finds relevant datasets that share common columns with the input base dataset to extend it with additional tuples. Basic datasets discovery usually evaluates each candidate dataset individually, emphasizing the similarity or overlaps between individual datasets and given keywords or base datasets. In contrast, our datasets assemblage evaluates a set of candidate datasets as a whole, emphasizing minimal overlaps with the input base dataset and among the datasets in the results. Moreover, basic datasets discovery is usually followed by a schema alignment step that aligns the schemas of the returned datasets with the input base dataset, e.g., when materializing the final dataset through union operations in table union search [19]. Our dataset assemblage builds upon the results of basic datasets discovery after schema alignment.

*Advanced datasets assemblage* allows users to specify fine-grained information needs to assemble the most desirable datasets that are evaluated as a whole. A recent study [36] defines a user's fine-grained information needs as specific data attributes. They focus on the richness of features in the datasets discovered. However, this can result in insufficient data instances and redundant features during model training, which may increase the risk of overfitting [75]. In contrast, our defined distinctiveness measure (see Definition 2.1) evaluates the distinctiveness of datasets, thereby facilitating the discovery of a more varied range of data instances.

**Tuples discovery** [7, 11, 21, 30, 37, 41, 73] is the process of selecting the most beneficial tuples from a pool of datasets, for a specific target such as model training [7], causal inference in question answering [21] or revenue allocation [43]. The "usefulness" of the tuples added is typically assessed in relation to a single target. However, our work differs in that it focuses on acquiring complete datasets (with less overlapping tuples) rather than individual tuples.

**Cardinality estimation**. As discussed in §1, estimating the distinctiveness for a set of datasets w.r.t. a query set can be cast as the multi-query-dataset cardinality estimation (MCE) problem. Therefore, we also examine existing research works for the single-query-dataset cardinality estimation (SCE) problem, which is generally divided into two categories: query-driven or data-driven [24, 32]. Query-driven approaches [25, 47, 54, 57, 62, 65, 67] train SCE models on historical queries. Conversely, data-driven approaches [26, 40, 64, 69, 71, 72, 76] train SCE models based on data distributions, without relying on any information from query workloads. While query-driven methods often lack flexibility, especially in the absence of representative queries, data-driven methods generally outperform query-driven methods [24]. Moreover, some hybrid methods [18, 33, 35, 42, 52, 68, 74] train SCE models by utilizing both data distributions and query workloads, which exhibit higher estimation accuracy and generality [24]. Despite the large body of existing works, current methods for SCE are not amenable to the MCE problem directly since they estimate cardinality for only a single query on a single dataset, whereas MCE requires estimating the cardinality for a query set on a set of datasets. This distinction underscores the key challenge in the MCE problem, which is to effectively identify overlaps among the tuple sets returned by each query in a query set across multiple datasets.

**Data pricing**. A recent survey [56] reports several data pricing functions. Commonly used pricing functions, such as tuple-based pricing [3, 37, 46], usage-based pricing [44, 45] and query-based pricing [10, 16, 17, 36, 38], primarily depend on the number of tuples in a dataset. Please see Appendix G for the detailed description. Without loss of generality, in our experiments, we adopt tuple-based pricing for datasets, as described in §E.1.

## 6 Conclusion

We introduced the problem of maximizing distinctiveness, which requires a subset of candidate datasets to be found that have the highest distinctiveness for a user-provided query set, a base dataset, and a budget. We first establish the NP-hardness of this problem. To solve this problem, we propose a greedy algorithm using ML-based distinctiveness estimation. This ML-based distinctiveness estimation method can effectively approximate the distinctiveness marginal gain without examining every tuple in each dataset. Using a comprehensive empirical validation on five real-world data pools, we demonstrate that our greedy algorithm using ML-based distinctiveness estimation is effective, efficient, and scalable.

Distinctiveness Maximization in Datasets Assemblage

Conference acronym 'XX, June 03–05, 2018, Woodstock, NY

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

## A  An Running Example

*Example A.1.* Suppose a user wishes to purchase datasets about Melbourne house information. As shown in Fig. 11, a user must spend \$400 to acquire a set of datasets, $D = \{d_1, d_5, d_{10}\}$, which are discovered using a base dataset $d_u$ at Stage 2(A) within the existing pipeline. Conversely, at Stage 2(B) in our pipeline, the user provides a query set $Q = \{q_1, q_2\}$ locating Melbourne house information. For each dataset $d \in D$, the union $Q(d)$ of tuple sets returned by each query in $Q$ on $d$ is obtained. A subset $S = \{d_5, d_{10}\}$ of $D$ exhibits maximum distinctiveness ($Q(d_5)$, $Q(d_{10})$, and $Q(d_u)$ each have two tuples. $Q(d_5)$ has an overlap with both $Q(d_{10})$ and $Q(d_u)$, resulting in the distinctiveness of $S$ of 4, i.e., $|Q(d_5) \cup Q(d_{10}) \cup Q(d_u)| = 4$). The user can purchase this subset for only \$300. Using our pipeline, a user spends less money while achieving the same tuples discovery results as $D$, as shown in Stage 3 of Fig 11.

## B  Hardness Analysis of the DM Problem

THEOREM B.1. *The DM problem is NP-hard.*

PROOF. We demonstrate a polynomial-time reduction from any instance of the MC problem to an instance of the DM problem. Here every element $e \in V$ from the MC instance is converted into a tuple. For categorical elements, each is assigned a unique index, which is then mapped to a tuple. A set of elements $S' \subseteq V$ in the MC problem corresponds to a dataset $d \in D$ in the DM problem. Hence, the universe $V$ is equivalent to the universe of all tuples $T_D = \bigcup_{d \in D} d$. By setting $p(d) = 1$, we map $K$ to $B$. Next, we define an empty base

---

**Algorithm 2** Exact-Greedy

**Input:** a set $D$ of datasets, a base dataset $d_u$, a query set $Q$, a budget $B$;
**Output:** a subset $S \subseteq D$ of datasets with distinctiveness;
1: $S \leftarrow \emptyset, \mathcal{T} \leftarrow \emptyset, T_S \leftarrow \emptyset$;
2: **for** $d \in D \cup d_u$ **do**
3:    $Q(d) \leftarrow \emptyset$;
4:    **for** $q \in Q$ **do** $q(d) \leftarrow$ ExecuteQueries$(d, q), Q(d) \leftarrow Q(d) \cup q(d)$;
5:    $\mathcal{T}[d] \leftarrow Q(d)$;
6: $T_S \leftarrow T_S \cup \mathcal{T}[d_u]$;
7: **while** $D \neq \emptyset$ **do**
8:    $g^* \leftarrow 0, d^* \leftarrow \emptyset, T^* \leftarrow \emptyset$;
9:    **for** $d \in D$ **do**
10:      $T \leftarrow T_S \cup \mathcal{T}[d], g \leftarrow (|T| - |T_S|)/p(d)$;
11:      **if** $g > g^*$ **then** $g^* \leftarrow g, d^* \leftarrow d, T^* \leftarrow T$;
12:    **if** $p(S) + p(d^*) \leq B$ **then** $S \leftarrow S \cup d^*, T_S \leftarrow T^*$;
13:    $D \leftarrow D \setminus d^*$;
14: $d_t \leftarrow \arg\max_{d \in D \wedge p(d) \leq B} |\mathcal{T}[d] \cup \mathcal{T}[d_u]|$;
15: **if** $|T_S| < |\mathcal{T}[d_t] \cup \mathcal{T}[d_u]|$ **return** $\{d_t\}$ and $|\mathcal{T}[d_t] \cup \mathcal{T}[d_u]|$;
16: **return** $S$ and $|T_S|$;

---

dataset $d_u = \emptyset$ and a query set $Q = \{q\}$ with $q = $ SELECT * FROM $d$ WHERE $min(T_D) \leq c \leq max(T_D)$ where $min(T_D)$ and $max(T_D)$ are the minimum and maximum values in $T_D$. As a result, $Q(d)$ includes all tuples in $d$, establishing a one-to-one correspondence between $S'$ and $Q(d)$. So, the objective of maximizing $|\cup_{S' \in \mathcal{S}} S'|$ and $|\bigcup_{d \in S \cup d_u} Q(d)|$ are equivalent. Therefore, the optimal solution of the DM problem also solves the optimal MC problem. Given the polynomial time complexity for this reduction, and the NP-hardness of MC, the DM problem is also NP-hard.  □

## C  Exact-Greedy

The pseudo-code of Exact-Greedy is shown in Alg. 2. Specifically, we begin by executing each query $q \in Q$ on every dataset $d \in D$, and on the base dataset $d_u$, followed by merging the tuple sets $q(d)$ returned for $q \in Q$ to obtain the union $Q(d)$ of all $q(d)$ (Line 4). We record $Q(d)$ in $\mathcal{T}[d]$ (Line 6). Subsequently, in each iteration, we add the dataset $d^*$ with the greatest marginal gain $g^*$ into $S$ until the budget is exhausted (Lines 7-13). In Line 14, we return a single dataset $d_t$ with the maximum distinctiveness $|\mathcal{T}[d_t] \cup \mathcal{T}[d_u]|$. Finally, we select the one with a larger distinctiveness from a set $S$ of datasets and a single best dataset $d_t$ (Lines 15-16). We present the associated approximation guarantee in Theorem C.1.

THEOREM C.1. Exact-Greedy *achieves an approximation ratio of $\frac{1-1/e}{2}$ when solving the DM problem.*

PROOF. Let $S^*$ be the optimal set of datasets, $S_m$ the set of datasets with size $m$ added to $S$ in the first $l$ iterations, $d_{m+1}$ the first dataset considered by $S^*$ but not included in $S$ since it exceeds the budget and $S_{m+1} = \{d_{m+1}\} \cup S_m$. Using Lemma 2 of [31], observe that $\mathscr{D}(S_{m+1}, d_u, Q) \geq (1 - \frac{1}{e})\mathscr{D}(S^*, d_u, Q)$ in Alg. 2. Let $\Delta\mathscr{D}$ be the increase in distinctiveness by including $d_{m+1}$ in $S_m$, $\mathscr{D}(S_{m+1}, d_u, Q) = \mathscr{D}(S_m, d_u, Q) + \Delta\mathscr{D} \geq (1 - \frac{1}{e})\mathscr{D}(S^*, d_u, Q)$. Since $\Delta\mathscr{D}$ cannot be greater than $\mathscr{D}(\{d_t\}, d_u, Q)$ for the best dataset $d_t$, $\mathscr{D}(S_m, d_u, Q) + \mathscr{D}(\{d_t\}, d_u, Q) \geq \mathscr{D}(S_m, d_u, Q) + \Delta\mathscr{D} \geq (1 - \frac{1}{e})\mathscr{D}(S^*, d_u, Q)$. Therefore, either $\mathscr{D}(S_m, d_u, Q)$ or $\mathscr{D}(\{d_t\}, d_u, Q)$ is greater than $\frac{(1-1/e)}{2}\mathscr{D}(S^*, d_u, Q)$, Exact-Greedy achieves an approximation ratio of $\frac{(1-1/e)}{2}$.  □

**Time complexity analysis**. Assume that $d$ is the dataset in $D$ with the greatest number of tuples. Alg. 2 requires $O(|d||Q|)$ time to execute each query $q \in Q$ and $O(|d|^2|Q|)$ time to combine the

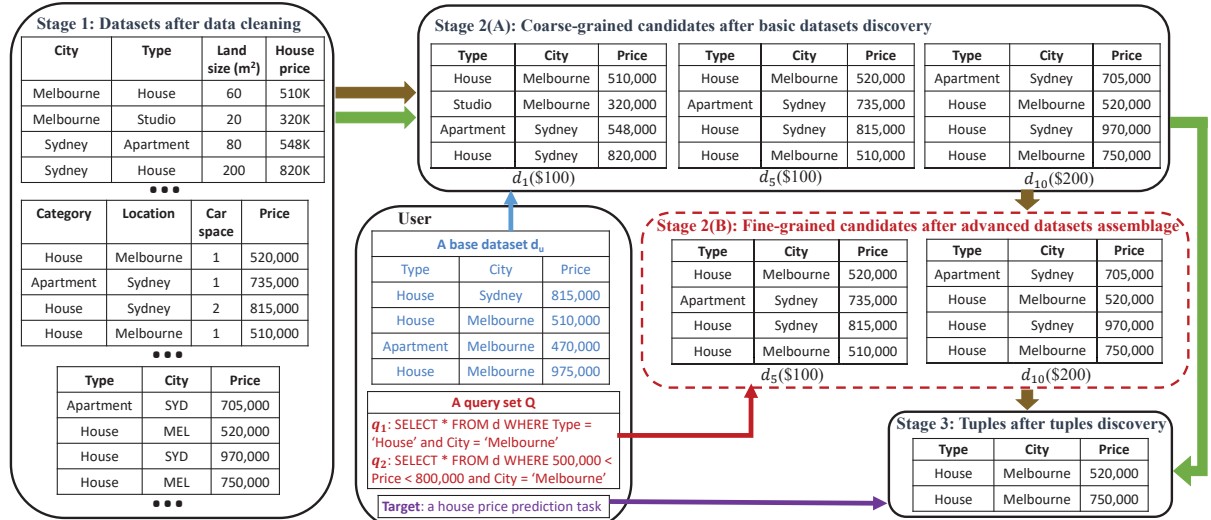

**Figure 11: An example of Fig. 1 where our pipeline achieves the same tuples discovery with a lower budget.**

---

**Algorithm 3** Distinctiveness($Q, E_C^d, n, C_Q$)

**Input:** the data summary $E_C^d$ of $d$, a query set $Q$, the number of tuples $n$ in $d$, a lookup table $C_Q$ to maintain column sets of each query $q \in Q$;

**Output:** estimated distinctiveness $\mathscr{D}$;

1: $\mathscr{D} \leftarrow 0, E_d \leftarrow \emptyset, E_Q \leftarrow \emptyset$
2: **for** $q \in Q$ **do**
3:   $e_d^q \leftarrow \emptyset, e_q \leftarrow \emptyset$;
4:   **for** $e_C \in E_C^d$ and $C \in C_Q[q]$ **do**
5:    $e_d^q \leftarrow [e_d^q, e_C], e_q \leftarrow [e_q, \psi(q_l^C), \psi(q_h^C)]$;
6:   $E_d \leftarrow E_d \cup e_d^q, E_Q \leftarrow E_Q \cup e_q$;
7: Generate $e_d^Q$ by each $e_d^q \in E_d$ using Eq. 4;
8: Generate $e_Q$ by $e_q \in E_Q$ using Eq. 5;
9: $\mathscr{D} \leftarrow \mathscr{D}(e_d^Q, e_Q) \times n$ using Eq. 6;
10: **return** $\mathscr{D}$;

---

**Algorithm 4** MergeEmbeddings($E_C^d, E_C^S, C_Q$)

**Input:** data summary $E_C^d$ of the dataset $d$, data summary $E_C^S$ of the set $S$ of datasets, and a lookup table $C_Q$ to maintain column sets associated with each query $q \in Q$;

**Output:** data summaries $E_C^{S \cup d}$;

1: $E_C^{S \cup d} \leftarrow \emptyset$;
2: **for** $e_C \in E_C^S$ **do**
3:   **if** $C \in \cup_{q \in Q} C_Q[q]$ **then**
4:    Find column set embedding $e'_C$ of $C$ from $E_C^d$;
5:    $e \leftarrow \delta(\phi_e([e_C, e'_C])), e_C \leftarrow e$;
6:   $E_C^{S \cup d} \leftarrow E_C^{S \cup d} \cup e_C$
7: **return** $E_C^{S \cup d}$;

---

tuple sets $q(d)$ for each query $q \in Q$ (Line 4). Therefore, constructing $\mathcal{T}[d]$ requires $O(|d|^2|Q||D|)$ time (Lines 2-5). Computing the marginal gain for the dataset $d$ requires $O(|d|^2|D|)$ time. So, finding the set $S$ of datasets with the maximum distinctiveness requires $O(|d|^2|D|^3)$ time (lines 7-13), resulting in a total time complexity for Alg. 2 of $O(|d|^2|Q||D| + |d|^2|D|^3)$.

## D  ML-Greedy

The pseudo-code of distinctiveness estimation is shown in Alg. 3. The procedure for merging data summaries is shown in Alg 4. The pseudo-code of ML-Greedy is shown in Alg. 5.

---

**Algorithm 5** ML-Greedy

**Input:** a set $D$ of datasets, a base dataset $d_u$, a query set $Q$, a budget $B$;

**Output:** a set $S \subseteq D$ of datasets with its distinctiveness;

1: Generate a set $C_d$ of column sets and data summary $E_C^d$ for each $d \in D \cup d_u$; /* offline process */
2: $C_Q \leftarrow \emptyset$;
3: **for** $q \in Q$ **do** $C_Q[q] \leftarrow \{C | C \in \cap_{d \in D \cup d_u} C_d \text{ and } C \cap \text{ColsOf}(q)\}$;
4: $S \leftarrow \emptyset, \mathbb{D} \leftarrow \emptyset, E_C^S \leftarrow \emptyset, n^* \leftarrow 0, \mathscr{D}^* \leftarrow 0$;
5: $\mathscr{D}^* \leftarrow$ Distinctiveness$(Q, E_C^{d_u}, |d_u|, C_Q), E_C^S \leftarrow E_C^{d_u}, n^* \leftarrow |d_u|$;
6: **while** $D \neq \emptyset$ **do**
7:   $g^* \leftarrow 0, d^* \leftarrow \emptyset, E^* \leftarrow \emptyset$;
8:   **for** $d \in D$ **do**
9:    $E_C^{S \cup d} \leftarrow$ MergeEmbeddings$(E_d, E_C^S, C_Q)$
10:    $\mathscr{D} \leftarrow$ Distinctiveness$(Q, E_C^{S \cup d}, n^* + |d|, C_Q), g \leftarrow \frac{\mathscr{D} - \mathscr{D}^*}{p(d)}$;
11:    **if** $S == \emptyset$ **then** $\mathbb{D}[d] \leftarrow \mathscr{D}$;
12:    **if** $g > g^*$ **then** $d^* \leftarrow d, g^* \leftarrow g, E^* \leftarrow E_C^{S \cup d}$;
13:   **if** $p(d^*) + p(S) \leq B$ **then**
14:    $E_C^S \leftarrow E^*, S \leftarrow S \cup d^*, \mathscr{D}^* \leftarrow \mathscr{D}^* + g^* \times p(d^*), n^* \leftarrow n^* + |d^*|$;
15:   $D \leftarrow D \setminus d^*$;
16: $d_t \leftarrow \arg\max_{d \in D \wedge p(d) \leq B} \mathbb{D}[d]$;
17: **If** $\mathscr{D}^* < \mathbb{D}[d_t]$ **return** $\{d_t\}$ and $\mathbb{D}[d_t]$;
18: **return** $S$ and $\mathscr{D}^*$;

---

**Time complexity analysis**. The proposed algorithm requires $O(|C_Q[q]||Q|)$ time to create the dataset and query embeddings for each query $q \in Q$ (Lines 2-6 in Alg. 3) and $O(8\eta^2 x^2|Q|)$ time to generate the corresponding query-aware dataset and query set embeddings (Lines 7-8 of Alg. 3). Thus, a total of $O(|C_Q[q]||Q| + 8\eta^2 x^2|Q|)$ time is required to estimate the distinctiveness for each dataset (Lines 1-9 in Alg. 3). Then, $O(\eta^2| \cup_{q \in Q} C_Q[q]|)$ time is required to merge the data summary $E_C^d$ of $d$ with the data summary $E_C^S$ of $S$ (Lines 1-6 in Alg. 4). Therefore, $O(|C_Q[q]||Q| + 8\eta^2 x^2|Q| + \eta^2| \cup_{q \in Q} C_Q[q]|)$ time is needed to compute the marginal gain for a dataset $d$ w.r.t. $S$ (Line 10 in Alg. 5). The total time complexity for the proposed algorithm that utilizes ML-based distinctiveness estimation is therefore $O((8\eta^2 x^2|Q| + \eta^2| \cup_{q \in Q} C_Q[q]|)|D|^2)$.

Note that in model pre-training, we set $\eta = 128$ (column set embedding size) and $x = 4$ (i.e., a query corresponds to at most 4 column sets). Moreover, $|d|$ is generally in the millions. Hence, the empirical efficiency improvement provided by ML-Greedy is

**Table 3: Data pools that are not used in model pre-training.**

| Name | # of columns (R/C)* | # of records |
|---|---|---|
| TPCH-LineItem [4] | 11/4 | 6M |
| DMV [60] | 7/13 | 11M |
| IMDB-CastInfo [15] | 6/1 | 36M |
| Airline-OnTime [14] | 66/17 | 440K |
| RealEstate [34] | 21/14 | 1.4M |

**Table 4: Parameter settings (default value in bold).**

| Parameter | Value |
|---|---|
| sample rate $s_{max}$ | **0.1**, 0.2, 0.3, 0.4, 0.5 |
| # of datasets $\|D\|$ | 10, **20**, 40, 80, 160 |
| overlap ratio $ol$ | 1%, 3%, **5%**, 7%, 9% |
| # of queries $\|Q\|$ | 10, **20**, 40, 80, 160 |
| budget $B$-ratio | 0.1, 0.3, **0.5**, 0.7, 0.9 |

**Table 5: Datasets used for model pretraining.**

| Name | # cols (R/C)* | Name | # cols (R/C)* | Name | # cols (R/C)* |
|---|---|---|---|---|---|
| Higgs | 28/0 | KDD99 | 34/5 | SUSY | 19/0 |
| PRSA | 16/2 | Gasmeth | 18/0 | Retail | 5/3 |
| Gastemp | 20/0 | Covtype | 10/9 | hepmass | 29/0 |
| Sgemm | 10/4 | Weather | 7/0 | Adult | 6/8 |
| PAMPA2 | 54/0 | YearPred | 90/0 | HTsensor | 10/0 |
| Power | 7/0 | WECs | 49/0 | Census90 | 0/50 |
| GasCO | 18/0 | | | | |

potentially even more substantial. This belief is verified in our experiments where we observe that ML-Greedy can be up to four orders of magnitude faster than Exact-Greedy.

## E  Preparing the Datasets and Queries

### E.1  Datasets Preparation

To control overlapping tuples in the candidate datasets, following existing related studies [49, 63], we generate each set of candidate datasets, $D$, by sampling tuples from a data pool, $d_{pool}$ (see Table 3 for five data pools). We introduce two new parameters, $s_{min}$ and $s_{max}$, which set the lower and upper bounds ($s_{min}\|D\|$, and $s_{max}\|D\|$) which is the expected number of tuples in $D$. We randomly choose a sampling rate $s$ within $[s_{min}, s_{max}]$ and then sample $s\|d_{pool}\|$ tuples from $d_{pool}$ to produce each candidate dataset $d$. We repeat this process to generate all datasets in $D$.

In our experiments, we set the size of $D$ to $\|D\| = 20$ by default. Additionally, we set $s_{min} = 1/\|D\| = 0.05$, implying that each tuple is expected to *appear in at least one dataset*. We also set $s_{max} = 2/\|D\| = 0.1$, indicating that each tuple is expected to *appear in at most two datasets*. This ensures that the datasets in $D$ have a realistic overlap of tuples but also maintain sufficient mutual distinctness.

**Pricing and budget**. For simplicity, we set $p(d) = w \times \|d\|$ for each dataset $d$ where $w$ is selected randomly from $(0, 1]$. This approach follows existing pricing functions [3, 37, 46], where prices are based on the number of tuples in datasets. To better control budget variations, we introduce the $B$-ratio, which is the ratio of the budget $B$ and the total price of the datasets in $D$, $\sum_{d \in D} p(d)$, and set it to the default value of 0.5. Note that data pricing is orthogonal to this work and alternative data pricing functions are discussed in Appendix G.

---

*R indicates real-valued columns and C indicates categorical columns.

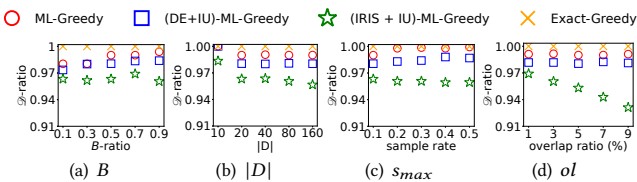

**Figure 12: The impact of varying $B$, $\|D\|$, $s_{max}$, $ol$ on the distinctiveness ratio of each algorithm for TPCH.**

### E.2  Query Set Preparation

To control overlapping tuples in the tuple sets returned by different queries, we introduce the parameter $ol$ which controls the *minimum overlap ratio* in the tuples returned by any pair of queries for a dataset. This idea is inspired by the approach used to generate queries in prior work [42]. To simplify the final configuration, each query is limited to using most one categorical column. Query set generation is derived from the number of available categorical columns, $k_c$, which is randomly set to either 0 or 1:

- For $k_c = 1$, we randomly select a categorical column $c$ from datasets in $D$. We set a value $v_c$ for $c$ by sampling $c$ from datasets in $D$, so that, in any dataset $d \in D$, the total number of tuples that satisfy $c = v_c$ exceeds $ol \times \|d\|$. We merge tuples that satisfy $c = v_c$ across datasets in $D$, to produce a new dataset $d_{sp}$. Next, we randomly select $k_r$ ($k_r \in [1, 3]$) real-valued columns from $d_{sp}$, setting the range to $c \in [\min(c), \max(c)]$, where $\min(c)$ and $\max(c)$ are the minimum and maximum values for $c$ in $d_{sp}$.
- For $k_c = 0$, we randomly sample $ol \times \|d\|$ tuples from each dataset $d \in D$ and merge them into the new dataset $d_{sp}$. We then randomly select $k_r$ ($k_r \in [2, 4]$) real-valued columns from $d_{sp}$, setting the range to $c \in [\min(c), \max(c)]$.

After generating the query $q$ using the above method, we generate another query $q'$ by selecting real-valued columns from the same sampled dataset $d_{sp}$ produced from $q$. The query pair $(q, q')$ must satisfy having a minimum overlap ratio $ol$ for the tuples returned from each dataset $d \in D$. This query pair is then added to the query pool. The query generation process terminates once a query pool of 100 query pairs is created. Query pairs are then randomly selected from the query pool to form a query set $Q$ and used with $D$.

By default, the number of queries $\|Q\| = 20$. Depending on the size of $D$, $ol$ is set to 5% by default to ensure the size of the sampled dataset $d_{sp}$ does not exceed the total number of tuples in any dataset $d \in D$, i.e., $\|d_{sp}\| \le \|D\| \times ol \times \|d\| \le \|d\|$, $ol \le 0.05$. This prevents a corner case where a query returns all tuples from the dataset.

## F  Supplementary Details for Experiments

### F.1  Implementation

**Parameter settings**. The key parameters introduced in Appendix E are summarized in Table 4. Three parameters govern the generation of data summaries: $\xi$ (the maximum number of bits per column), $\ell$ (the number of bits per row), $\eta$ (the column set embedding size). The same parameters are used by IRIS [42]. We set $\xi = 128$, $\ell = 2048$, and $\eta = 128$ for DE and IRIS, which are also the default values in prior work [42]. Note that, DE has an additional parameter, $x$, which is the maximum number of column sets that a query corresponds

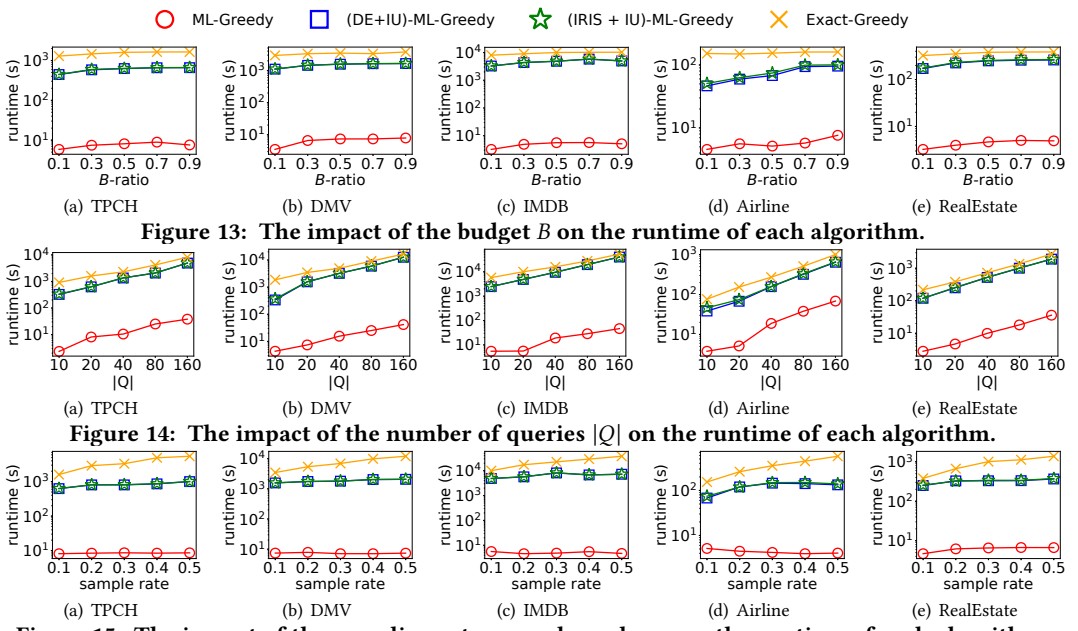

Figure 13: The impact of the budget $B$ on the runtime of each algorithm.

Figure 14: The impact of the number of queries $|Q|$ on the runtime of each algorithm.

Figure 15: The impact of the sampling rate upper bound $s_{\max}$ on the runtime of each algorithm.

to during the generation of a query-aware dataset embedding and a query-set embedding. We set this parameter to $x = 4$.

**Training set generation**. We pretrain our models using the 19 datasets listed in Table 5. These datasets are publicly available [42] and are used to pretrain the SCE models.

To train the models for distinctiveness estimation, we follow the approach used in prior work [42]. We select 5 column sets with the highest correlation score, as calculated using CORDS [28], from the training datasets and 300 randomly generated queries. For each column set, we record the top-10 queries by cardinality. We combine every two column sets from the same training dataset to generate the query pairs, resulting in 100 query pairs, for any two column pairs, and create a total of 19,000 query pairs. We allocate 80% of the query pairs for training and 20% for validation, and use a batch size of 256. To train the model that is used to merge data summaries, we randomly select 10,000 column pairs from the 19 datasets. For each training dataset, we sample two datasets, $d_1$ and $d_2$, with a sample rate of 0.5. For each column pair in the training dataset, we use Eq. 2 to compute the column set embeddings $e_1$ and $e_2$ on $d_1$ and $d_2$, respectively. In addition, we generate a column set embedding $e$ for the dataset resulting from the merge of $d_1$ and $d_2$ using Eq. 2.

**Experimental environment** We perform all experiments on a server running Red Hat Enterprise Linux with an Intel® Xeon® CPU@2.60GHz, 512GB of memory, and two Nvidia Tesla P100 GPUs, each with 16GB of memory. We implement all algorithms in Python.

## F.2 Additional Effectiveness Results

Fig. 12 shows, for TPCH, the impact of varying the budget $B$, the number of datasets $|D|$, the sampling rate upper bound $s_{max}$, and the minimal overlap ratio $ol$ between query pairs on the distinctiveness ratio of the considered algorithms. They have a similar trend with those results for other datasets in Figs. 5-8.

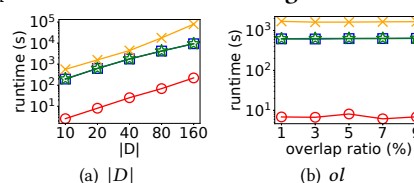

Figure 16: The impact of varying $|D|$ and $ol$ on the runtime of each algorithm for TPCH.

## F.3 Additional Efficiency Results

**Impact of the budget $B$.** The impact of varying the budget ratio is shown in Figs. 13(a)-13(e). Observe that: (1) As the $B$-ratio is increased, more datasets are selected, and the runtime for all of the algorithms increases. However, ML-Greedy is at least two orders of magnitude faster than all other algorithms. (2) Our distinctiveness estimation method DE is an order of magnitude faster than the exact distinctiveness computation, and is three times faster than IRIS. (3) Although (DE+IU)-ML-Greedy uses the same distinctiveness estimation method as ML-Greedy, it is an order of magnitude slower than ML-Greedy. This demonstrates the high efficiency of our approach, MS, to update data summaries.

**Impact of the number of queries $|Q|$.** Based on Figs. 14(a)-14(e), observe that: (1) With increasing $|Q|$, the runtime for all algorithms increases as more queries must be processed when estimating distinctiveness. However, our algorithms are at least two orders of magnitude faster than the other algorithms. (2) Our distinctiveness estimation method DE is two orders of magnitude faster than the exact distinctiveness computation, and is two times faster than IRIS. (3) Our approach to updating data summaries, MS, is at least an order of magnitude faster than IU.

**Impact of the sampling rate upper bound $s_{\max}$.** From Figs. 15(a)-15(e), we observe: (1) For larger values of $s_{\max}$, our approach is at least two orders of magnitude faster than the baselines. (2) As $s_{\max}$ increases, the runtime of Exact-Greedy increases since it takes

more time to execute the queries and compute the marginal gain for more tuples that are returned by the queries. All other algorithms appear to be insensitive to changes to $s_{max}$. This demonstrates the advantage of using our distinctiveness estimation method, i.e., effectively eliminating the need to test each tuple individually. (3) Our distinctiveness estimation method DE is three orders of magnitude faster than the exact distinctiveness computation, and three times faster than IRIS. (4) Our method for updating data summaries, MS, is two orders of magnitude faster than IU.

**Additional efficiency results for TPCH**. Fig. 16 shows, for TPCH, the impact of varying the number of datasets $|D|$ and the minimal overlap ratio $ol$ between query pairs on the runtime of the considered algorithms. They have a similar trend with those results for other datasets in Figs. 9-10.

# G Supplementary Details for Related Work

**Data pricing**. Tuple-based pricing functions [3, 37, 46] assign a price to each tuple, and the price of a dataset is the sum of the tuple prices. Usage-based pricing functions [44, 45] charge based on the dataset's usage, measured in bytes transferred per API request, with the data transferred linked to the dataset's tuple count. Query-based pricing functions [10, 16, 17, 36, 38] charge for the query results returned from a dataset rather than providing the entire dataset, with the price of query results also depending on the number of tuples returned.

