# OpenReview forum: "Distinctiveness Maximization in Datasets Assemblage"
_ACM.org/TheWebConf/2025/Conference — WWW 2025 Poster_

### Official Review · Reviewer_zzxs · 2024-11-21

**Novelty:** 4
**Technical Quality:** 4

**Review:**

The paper presents a novel approach to dataset assemblage aimed at maximizing distinctiveness under a given budget. It introduces an ML-based method to estimate distinctiveness marginal gain for candidate datasets, addressing the limitations of exact computation methods. The approach integrates components like query-aware embeddings and distinctiveness estimation, significantly improving scalability and efficiency. The authors demonstrate the effectiveness of their algorithm through experiments on real-world datasets and validate its utility for downstream machine learning tasks.

Weakness:

1. The layout of the article is overly compressed, making it somewhat hard to follow.

**Questions:**

1. How does the performance of the proposed method scale with the complexity of queries (not the number of queries) in the query set? For instance, does the method handle highly nested SQL queries or queries with complex conditions effectively?
2. Please provide a breakdown of computational resource requirements for the ML-based distinctiveness estimation components, particularly during training and online inference phases.
3. Have you considered integrating dynamic updates to the dataset pool? If so, how would the method adapt to continuously incoming datasets or evolving query sets?
4. It would be better if the authors expand the range of baselines to include additional heuristic and non-ML-based methods, offering a more comprehensive comparison.

**Reviewer Confidence:**

2: The reviewer is willing to defend the evaluation, but it is likely that the reviewer did not understand parts of the paper

**Scope:**

3: The work is somewhat relevant to the Web and to the track, and is of narrow interest to a sub-community

---

### Official Review · Reviewer_nHk5 · 2024-11-29

**Novelty:** 3
**Technical Quality:** 3

**Review:**

The paper introduces a novel approach to dataset assemblage that aims to maximize the number of distinct tuples within a given budget. The authors define a problem of distinctiveness maximization and propose a greedy algorithm with machine learning (ML)-based distinctiveness estimation to approximate the solution efficiently. The manuscript has been evaluated against five real-world data pools, demonstrating the effectiveness, efficiency, and scalability of the proposed method.

However, I have several concerns that need to be addressed.

1. The problem is not so interesting. The paper could benefit from a clearer articulation of the practical use cases for the problem of distinctiveness maximization.

2. The introduction section of the paper is somewhat tedious and lacks a compelling narrative.

3. It seems to me that the authors complicate the problem by coining the concept of Distinctiveness. In my view, it would be sufficient to use existing descriptions such as "cardinality estimation."

4. The paper lacks information regarding the training of the ML model used for distinctiveness estimation. The authors should provide details on the training set, the model architecture, and the training process. This information is crucial for reproducibility and for assessing the validity of the proposed method.

5. The method requires users to provide a query, which may be too restrictive in practice. Often, users are unaware of the available data and cannot formulate precise queries, or they may be interested in datasets that do not align with their initial expectations.

6. In the running example provided in Figure 11, if the user is interested in Melbourne's housing information, it seems logical to first filter out irrelevant information, such as Sydney's housing data, from the base dataset.

7. The experiments in the paper primarily utilize data-centric datasets. However, many datasets consist largely of text, and the performance of the proposed method on text-centric datasets should be validated. The authors should consider whether their approach can be extended or adapted to handle datasets with a significant text component.

**Questions:**

1. Could you provide more specific examples or case studies where the problem of distinctiveness maximization is particularly relevant, and how it differs from or is superior to existing methods in these contexts?

2. Could you detail the training process for the ML model used in distinctiveness estimation?

3. How does your method perform with text-centric datasets, which are common in many real-world applications? Have you conducted any experiments to evaluate the effectiveness of your approach on such datasets?

**Reviewer Confidence:**

3: The reviewer is confident but not certain that the evaluation is correct

**Scope:**

4: The work is relevant to the Web and to the track, and is of broad interest to the community

---

### Official Review · Reviewer_S5oh · 2024-12-01

**Novelty:** 6
**Technical Quality:** 5

**Review:**

1. paper summary
This paper tackles the problem of assembling datasets that maximize distinctiveness, defined as the count of unique tuples across a collection of datasets, under budget constraints. The authors highlight the computational challenges of exact distinctiveness calculation, which requires evaluating the marginal gain of every dataset through exhaustive tuple-level comparisons. They prove this problem to be NP-hard and propose a machine learning (ML)-based solution to estimate distinctiveness efficiently without directly calculating overlaps for every query and dataset combination. By leveraging pre-trained embeddings and a series of novel algorithms, the proposed method significantly reduces computational overhead while maintaining high accuracy. Extensive experiments on five real-world datasets demonstrate that the method outperforms existing approaches in terms of both efficiency and effectiveness, with notable improvements in scalability. Furthermore, a case study on downstream machine learning tasks, including classification and regression, illustrates the practical utility of the approach in finding datasets that improve model performance. This work advances dataset assemblage methodology, combining theoretical rigor with practical innovation, to address a critical gap in data preparation pipelines.


2. Paper Strengths
1) Novel Problem Formulation: The paper extends traditional single-query cardinality estimation to multi-dataset, multi-query settings, which is both novel and impactful.
2) Efficiency: The proposed ML-based algorithm achieves significant runtime improvements compared to baseline methods while maintaining high accuracy.
3) Experimental Validation: Comprehensive experiments on five real-world datasets showcase the method's scalability and effectiveness in diverse scenarios.
4) Practical Utility: The method integrates seamlessly into data preparation pipelines and addresses real-world constraints like budget and information redundancy.


3. Paper Weaknesses
1) Limited Theoretical Details: The paper could elaborate more on the theoretical underpinnings of the ML-based estimation model, especially the approximation guarantees.
2) Use Case Generalization: While two downstream tasks are presented, the application scope beyond these tasks (e.g., in recommendation systems or data integration) is not explored in detail.

4. Detailed Comments
1) The explanation of the column set embedding generation process (Section 3.1) is clear, but additional examples could enhance understanding for a broader audience.
2) The impact of parameters like overlap ratio (Figure 8) is well-visualized, but more context on how these parameters relate to real-world datasets would be beneficial.
3) The comparisons with state-of-the-art methods (e.g., IRIS) are rigorous, but the paper could discuss scenarios where the baseline methods might outperform the proposed approach.

**Questions:**

The distinctiveness maximization problem addressed in this paper is closely related to cardinality estimation, where HyperLogLog and refined hyperloglog (Fine-grained probability counting for cardinality estimation of data streams) are well-known for their efficiency in estimating the cardinality of large sets with limited memory. Could the authors clarify why HyperLogLog or its extensions were not discussed or compared in this work? Additionally, given the complexity of multi-dataset-query cardinality estimation (MCE), do the authors see any potential to incorporate probabilistic data structures like HyperLogLog or refined hyperloglog into their framework for distinctiveness estimation?

**Reviewer Confidence:**

3: The reviewer is confident but not certain that the evaluation is correct

**Scope:**

4: The work is relevant to the Web and to the track, and is of broad interest to the community

---

### Official Review · Reviewer_VM4m · 2024-12-02

**Novelty:** 6
**Technical Quality:** 6

**Review:**

This paper addresses the challenge of assembling datasets to maximize distinctiveness from a collection, given a user's query set and budget constraints. The authors propose a novel machine learning-based method that efficiently estimates the distinctiveness marginal gains of datasets without the need for exhaustive tuple scanning. This approach has been evaluated on several datasets, which has been proven to improve the scalability and efficiency significantly.

Pros:

1.	This paper addresses a practical problem, improving data utility and efficiency in various real-world applications.
2.	This approach transforms the distinctiveness estimation problem into the multi-dataset-query cardinality estimation problem. And it further introduces an ML-based solution, significantly reducing the computational overhead.
3.	The enhanced experiments demonstrate superior performance against existing methods.

Cons:

1.	The scope of this paper and the problem to be addressed should be specifically and clearly described to better position the paper within the current research landscape. For example, data preparation is a complex problem due to various data formats, which means that not all datasets discovery practices contain stage 3 or need to select tuples.

**Questions:**

Q1: What are the main limitations of this approach? For example, what types of datasets can not be enriched by the proposed method?

Q2: Section 3 introduces many algorithm design details but lacks explanation of why it is designed this way.

**Reviewer Confidence:**

3: The reviewer is confident but not certain that the evaluation is correct

**Scope:**

4: The work is relevant to the Web and to the track, and is of broad interest to the community

---

### Official Review · Reviewer_XvJp · 2024-12-02

**Novelty:** 4
**Technical Quality:** 4

**Review:**

## Summary

This paper addresses the problem of assembling datasets with maximum distinctiveness under a budget constraint. The authors propose a novel ML-based method to estimate distinctiveness for dataset collections and develop a greedy algorithm (ML-Greedy) that achieves better efficiency than exact computation methods while maintaining competitive effectiveness. The work includes theoretical analysis proving NP-hardness and approximation guarantees, along with extensive experiments on real-world datasets.

## Strength

- Novel problem formulation for dataset assemblage with budget constraints
- Theoretical analysis with proven approximation bounds
- Practical ML-based solution that significantly improves efficiency

## Weakness

- Limited theoretical analysis of the ML-based estimation approach
- Lack of discussion on how the pricing function affects results
- No comparison with more recent dataset discovery methods
- Limited discussion of the solution's limitations

## Detailed Comments

[MAJOR] The paper appears to be out of scope for WWW. While data management and dataset discovery are important topics, the paper lacks significant connection to Web technologies, Web-scale data, or Web-specific challenges.

### Originality
The distinctiveness maximization problem formulation, while novel, seems to be a natural extension of existing dataset discovery problems. More discussion is needed on why previous approaches are insufficient.


### Importance of Contribution
The paper doesn't adequately justify why distinctiveness is the most appropriate metric for dataset usefulness. Alternative metrics should be discussed.
The connection between distinctiveness and actual data utility needs stronger empirical validation across more diverse use cases.

### Soundness
The ML model's architecture choices and training process are not well justified. Why were these specific choices made?
The paper lacks analysis of how the ML model's accuracy affects the final solution quality. What is the relationship between estimation errors and solution optimality?
The theoretical analysis of the greedy algorithm assumes exact computation. How do the bounds change with estimation errors?

### Evaluation
The experiments are limited to only five data pools. More diverse datasets would strengthen the evaluation.
The impact of query complexity on performance is not thoroughly evaluated. How does the solution scale with more complex queries?
Missing ablation studies on different components of the ML model.
No evaluation of the solution's robustness to different pricing models.
The case study on downstream ML tasks only covers classification and regression. Other important tasks like clustering or ranking should be considered.

**Questions:**

- Could you clarify how your approach specifically addresses Web-scale data challenges? While the paper presents a solution for dataset assemblage, it's not clear how it handles the unique characteristics of Web data such as heterogeneity, scale, and real-time updates that are crucial for WWW conference scope.
- In your ML-based distinctiveness estimation method, how robust is the model's performance when dealing with different data distributions? Specifically, if the test datasets have significantly different characteristics from the training datasets, how does this affect the estimation accuracy and the final solution quality? Some empirical evidence on this would help understand the method's generalizability.

**Reviewer Confidence:**

2: The reviewer is willing to defend the evaluation, but it is likely that the reviewer did not understand parts of the paper

**Scope:**

2: The connection to the Web is incidental, e.g., use of Web data or API